# Distributed Consensus Tracking Control of Chaotic Multi-Agent Supply Chain Network: A New Fault-Tolerant, Finite-Time, and Chatter-Free Approach

**DOI:** 10.3390/e24010033

**Published:** 2021-12-24

**Authors:** Ziyi Liu, Hadi Jahanshahi, Christos Volos, Stelios Bekiros, Shaobo He, Madini O. Alassafi, Adil M. Ahmad

**Affiliations:** 1School of Economics and Management, Hunan Open University, Changsha 410004, China; custard1006@163.com; 2Department of Mechanical Engineering, University of Manitoba, Winnipeg, MB R3T 5V6, Canada; jahanshahi.hadi90@gmail.com; 3Laboratory of Nonlinear Systems, Circuits & Complexity (LaNSCom), Department of Physics, Aristotle University of Thessaloniki, 54124 Thessaloniki, Greece; 4Department of Banking and Finance, FEMA, University of Malta, MSD 2080 Msida, Malta; stelios.bekiros@eui.eu; 5Department of Economics, European University Institute, I-50014 Florence, Italy; 6School of Physics and Electronics, Central South University, Changsha 410083, China; heshaobo@csu.edu.cn; 7Department of Information Technology, Faculty of Computing and Information Technology, King Abdulaziz University, Jeddah 21589, Saudi Arabia; malasafi@kau.edu.sa (M.O.A.); aahmad@kau.edu.sa (A.M.A.)

**Keywords:** consensus tracking, super-twisting sliding mode, supply chain network, faults in control signal, finite-time estimator

## Abstract

Over the last years, distributed consensus tracking control has received a lot of attention due to its benefits, such as low operational costs, high resilience, flexible scalability, and so on. However, control methods that do not consider faults in actuators and control agents are impractical in most systems. There is no research in the literature investigating the consensus tracking of supply chain networks subject to disturbances and faults in control input. Motivated by this, the current research studies the fault-tolerant, finite-time, and smooth consensus tracking problems for chaotic multi-agent supply chain networks subject to disturbances, uncertainties, and faults in actuators. The chaotic attractors of a supply chain network are shown, and its corresponding multi-agent system is presented. A new control technique is then proposed, which is suitable for distributed consensus tracking of nonlinear uncertain systems. In the proposed scheme, the effects of faults in control actuators and robustness against unknown time-varying disturbances are taken into account. The proposed technique also uses a finite-time super-twisting algorithm that avoids chattering in the system’s response and control input. Lastly, the multi-agent system is considered in the presence of disturbances and actuator faults, and the proposed scheme’s excellent performance is displayed through numerical simulations.

## 1. Introduction

In distributed coordination control, agents interact cooperatively through decentralized controllers using limited inter-agent communication and local information. Due to its wide range of applications, such as sensor networks, multi-robots, multiple economic systems and so forth, consensus plays a significant role in the research of distributed coordination control [1,2,3]. Although many studies on the consensus tracking problem have been published to date, most of them suffer from significant drawbacks, including vulnerability against disturbances and faults in control agents [4,5].

The objective of a supply chain is to satisfy consumer requirements in the most cost-effective way possible: in the right place, at the right time, and at the appropriate service level [6,7]. Supply chains are nonlinear dynamical systems that are driven by a variety of unknown stimulants such as customer demands [8]. Because of growing consumer expectations and strong competition in global marketplaces, businesses have always attempted to manage their supply chain networks to obtain the best possible results [9]. Although many variables, such as transportation fleets, machinery and equipment, play key roles in the performance of supply chains, the efficacy of such systems is largely determined by management decisions [10]. Nonetheless, uncertainties and faults in such systems make controlling and managing them difficult [11].

Academies and decision-makers have been working for decades to develop a dependable method for dealing with sophisticated supply chain networks [12]. In the past, decision-makers relied heavily on intuition and experience to handle their businesses. However, as time has passed, supply chains have grown increasingly complex, leading to the breakdown of traditional management approaches [13,14]. The supply chain’s business entities are entangled in a web of uncertainty. In addition, most supply chain networks have nonlinear dynamics, which can result in chaotic reactions [15,16]. As a result, using the most up-to-date procedures for their manganates is essential [17].

Sliding mode control (SMC) is the most popular robust controller and has garnered a lot of attention because of its features, including simplicity of implementation, assured stability and resilience to parameter changes [18,19,20]. Many studies on the application of sliding mode control in the control of chaotic systems have recently been carried out [21,22]. The SMC, on the other hand, has certain flaws. The conventional SMC, for example, does not guarantee convergence in a finite amount of time. In order to address the problem of finite-time convergence, the terminal sliding mode control (TSMC) was created [23,24].

Designing control and management techniques that ensure the systems’ appropriate performance, as previously noted, is a key issue in this area [25,26,27,28,29,30,31,32,33,34]. The strong foundation for dealing with nonlinear dynamics has been laid by control theory [35,36,37,38,39,40,41]. As a result, applying control theory to supply chain networks with nonlinear dynamics can provide impressive outcomes [42]. Over the previous half-century, several approaches established in the control literature have been applied to supply chains [43,44]. Despite the efforts of academics, this field of study still demands more attention [45,46]. For example, no research in the literature takes into account finite-time convergence and control input restrictions in supply chain control. The majority of financial systems have nonlinear dynamics that are sensitive to shocks and control input constraints [47,48]. As a result, several research studies have presented reliable methods for controlling economic and financial systems [49,50,51]. Nonlinear observers should be used to identify the dynamics of external disturbances because they cannot be directly evaluated in a nonlinear environment. On the other hand, faults and failures should be considered in supply chain networks. Nonetheless, there is no study in the literature to propose a fault-tolerant control scheme for supply chain networks.

The aforementioned concerns have prompted the current study. We propose a new fault-tolerant, finite-time, and chatter-free approach for distributed consensus tracking of chaotic multi-agent supply chain networks. The benefits of the proposed method in comparison with conventional sliding mode controller are listed in the following: In the real application, the existence of faults and disturbances are deniable. In the proposed controller, under the fault-tolerant approach, the limitation in the actuators is considered. The presented stability criteria assure the stability of the system in the presence of faults and disturbances;While we use most sliding mode controls, it is possible to see chattering in the response of the system, which is unfavorable. The super-twisting method is used to develop an effective controller without chattering;In comparison to traditional sliding mode control, the developed control scheme ensures system control in a finite time.

The rest of this paper is arranged as follows: in Section 2, the multi-agent supply chain network is presented, and its chaotic behavior is illustrated. Section 3 outlines the suggested control scheme’s design method, which considers impacts of disturbances and actuator faults using the finite-time disturbance-observer, and the super-twisting SMC. The Lyapunov stability theorem is also used to verify the closed-loop system’s stability in finite-time. In Section 4, the proposed control technique is applied to the multi-agent systems, and results of distributed consensus tracking are demonstrated. Lastly, concluding remarks and suggestions for future research are listed in Section 5.

## 2. Supply Chain Network and Its Corresponding Multi-Agent System

### 2.1. Chaotic Supply Chain Network

Many academics have attempted to simulate supply chain networks so far. The nonlinear model developed by Anne et al. [52] has received a lot of attention among all models proposed for supply chain networks. They developed a nonlinear supply chain model that takes into account safety stock, information distortion, and retailer order fulfillment. This model is provided by the following state-space equations:(1)$${\dot{y}}_{1}\left(t\right)\;=m\;y_{2}\left(t\right)\;-\;\left(n+1\right)y_{1}\left(t\right)\;{\dot{y}}_{2}\left(t\right)\;=r\;y_{1}\left(t\right)\;-\;y_{2}\left(t\right)\;-\;y_{1}\left(t\right)y_{3}\left(t\right)\;{\dot{y}}_{3}\left(t\right)\;=y_{1}\left(t\right)\;y_{2}\left(t\right)\;+\;\left(k-1\right)y_{3}\left(t\right),$$
where $y_{1}$, $y_{2}$, and $y_{3}$, respectively, stand for the current period’s quantity requested by the retailer, the amount of merchandise that distributors can deliver in the current period, and the current period’s quantity produced depending on the order. m also denotes the rate of customer demand satisfaction at a retailer. n indicates the distributors’ inventory levels. k stands for manufacturer’s safety stock coefficient, and r is the rate of product information distortion requested by retailers.

Figure 1 and Figure 2 demonstrate the chaotic attractor of the supply chain network (1) when the values of system parameters are set to $\left(m,\;n,\;r,\;k\right)=\left(12,\;7,\;45,-\frac{7}{3}\right).$ The following are the starting conditions for state variables: $\left(x_{1}\left(0\right),\;x_{2}\left(0\right),\;x_{3}\left(0\right)\right)=\left(4,-2,\;3\right)$. The nonlinear supply chain network exhibits chaotic behavior under this setting, as seen in these diagrams. It is noteworthy that, since we have used the non-dimensional model in this study, all numerical results are dimensionless.

**Definition** **1** **(weighed graph).**
*Suppose*

$G=\left(v,E\right)$

*as a weighted graph where*

$v=\left(v_{1}\;\cdot \cdot \cdot \;v_{n}\right)$

*stands for the nonempty set of nodes, and*

E ⊆v×v

*indicates the set of edge, while*

$\left(v_{i},v_{j}\right)\in E\;$

*means from node*

i

*to node*

j

*there is an edge. The topology of a weighted graph*

G

*indicates the adjacency matrix*

$\;A=\left[a_{ij}\right]\in R^{N\times N}\;$

*in which*

$a_{ij}>0\;$

*if*

$\left(v_{j},v_{i}\right)\;\in E$

*, otherwise*

$a_{ij}=0$

*. Weighted graph*

G

*is considered to be a direct graph. The node*

i

*’s weighted in-degree is defined as*

$d_{i}=\overset{N}{\underset{j=1}{\sum }}a_{ij}$

*; and consequently, the in-degree matrix is*

$D=diag\left(d_{1}\;\cdot \cdot \cdot \;d_{N}\;\right)\in R^{N\times N}.$

*The Laplacian matrix of a graph is*

$\;L=D-A\in \;R^{N\times N}$

*.*


**Definition** **2** **(directed graph).**
*In a directed graph, the set of neighbors of node *

i

*is all the nodes from which the node*

i

*may obtain information, not necessarily vice versa. Neighbor is a mutual relation in an undirected graph. A direct path between nodes*

i

*and*

j

*is a series of straight edges form*

$\left\{\left(v_{i},v_{l}\right)\;\left(v,v_{k}\right)\;\cdot \cdot \cdot \;\left(v_{m},v_{j}\right)\right\}$

*.*


### 2.2. Proposed Multi-Agent Three-Echelon Supply Chain

Because there are several analogies between a company in a business network and an agent, the multi-agent System paradigm can be a viable approach for modeling supply chain networks [53]. In 1998, for the first time, Lin et al. [54] proposed a multi-agent information system approach to model the order fulfillment process in supply chain networks. After that, many researchers have worked on supply chain systems from this point of view. In the current study, following this research flow, based on the supply chain that was modeled by Anne et al. [52], we propose a multi-agent supply chain network. The parameters of the model are considered based on reference [52].

In the current study, we offer a multi-agent system of three-echelon supply chain (1). Figure 3 shows the communication topology graph with one leader and four followers.

The leader and four agents are supposed to be supply chain networks with different initial conditions. The dynamic of the leader is considered as system (1). The dynamical models of follower agents j=1,2,3,4 are given by:(2)$${\dot{x}}_{1j}\left(t\right)\;=m_{j}\;x_{2j}\left(t\right)\;-\;\left(n_{j}+1\right)x_{1j}\left(t\right)\;+\;d_{1j}+u_{1j}\;{\dot{x}}_{2j}\left(t\right)\;=r_{j}\;x_{1j}\left(t\right)\;-\;x_{2j}\left(t\right)\;-\;x_{1j}\left(t\right)x_{3j}\left(t\right)\;+\;d_{2j}+u_{2j}\;{\dot{x}}_{3j}\left(t\right)\;=x_{1j}\left(t\right)\;x_{2j}\left(t\right)\;+\;\left(k_{j}-1\right)x_{3j}\left(t\right)\;+\;d_{3j}+u_{3j}.$$

In what follows, a new consensus tracking methodology is proposed and applied to the chaotic multi-agent supply chain networks.

## 3. Controller Design

### 3.1. Problem Formulation

Without losing generality, let the state space of the *i*th follower as below:(3)$${\dot{x}}_{i}\left(t\right)\;=f_{i}\left(x_{i}\right)\;+\;\Delta f_{i}\left(x_{i}\right)\;+\;\left(g_{i}\left(x_{i}\right)\;+\;\Delta g_{i}\left(x_{i}\right)\right)u_{i}+d_{i}\left(t\right),$$
with being $x_{i}={\left[x_{i1},\;x_{i2},\ldots ,\;x_{in}\right]}^{T},\;u_{i}={\left[u_{i1},\;u_{i2}\;,\ldots ,\;u_{in}\right]}^{T}$, and $d_{i}={\left[d_{i1},\;d_{i2}\;,\ldots ,\;d_{in}\right]}^{T}$ the state vector, control input, and disturbances, respectively. $f_{i}$ and $g_{i}$ denote nonlinear functions of the systems. Δf and Δg indicate the uncertainties and structural variations that there are in the dynamics of the system.

According to the definitions that are presented references in [55,56,57] faults and/or failures can be modelled in the following way:(4)$$\begin{array}{cc} u_{i}= & u_{ci}+b_{i}\;\left(t\right)\;\left(\left(e_{ii}-1\right)u_{ci}+{\bar{u}}_{i}\right)\\ i= & 1,\;2,\ldots ,\;n, \end{array}$$
in which the desired control input is shown by $u_{ci}$, the actual control input is represented by $u_{i}$, and ${\bar{u}}_{i}$ denotes the uncertain fault input. Parameter $0\;\leq e_{ii}\;\leq 1$ indicates the effectiveness of the control actuator. The time-varying function of a fault affecting the actuator is represented by:(5)$$b_{i}\left(t\right)\;=\;\left\{\begin{array}{c} 0,\;\;\;\;\;\;\;\;\;\;\;\;\;\;\;\;\;\;\;\;\;\;\;\;\;\;\;\;\;\;\;\;\;\;\;\;\;\;\;\;\;\;t<t_{0i}\\ 1-e^{-a_{i}\left(t-t_{0i}\right)}\;\;\;\;\;\;\;\;\;\;\;\;\;\;\;\;\;\;\;\;\;\;\;\;t\geq t_{0i} \end{array}\right.,$$
with $a_{i}>0$ as the unknown fault evolution rate, and $t_{0i}$ as the moment that the fault is started. Consequently, by considering faults and failures, the system’s control input is given by:(6)$$u=u_{c}+B\left(t\right)\left(\left(E\;-\;I\right)u_{c}+\bar{u}\right),$$
in which $E=diag\left(e_{11},\;e_{22},\ldots ,\;e_{nn}\right)$ denotes the effectiveness matrix. $B\left(t\right)=diag\left(b_{1}\left(t\right),\;b_{2}\left(t\right),\ldots ,\;b_{n}\left(t\right)\right)$ denotes the time profile of faults. In addition, the desired control input and additive fault output vector are respectively represented by $u_{c}={\left[u_{c1},\;u_{c2},\ldots ,\;u_{cn}\right]}^{T}$ and $\bar{u}={\left[{\bar{u}}_{1},\;{\bar{u}}_{2},\ldots ,\;{\bar{u}}_{n}\right]}^{T}$. Accordingly, we define the state space equation of system (3) with actuator faults and/or failures as follows:(7)$$\begin{array}{cc} \dot{x}\left(t\right)\;= & f\left(x\right)\;+\;d+u\\ u= & u_{c}+B\left(t\right)\left(\left(E-I\right)u_{c}+\bar{u}\right). \end{array}$$

**Assumption** **1.**
*Additive fault*

${\bar{u}}_{i}$

*is bounded, i.e.,*

$\left|{\bar{u}}_{1}\right|$

*≤*

$u_{0}$

*. In addition, control actions are limited due to the physical limits of the actuators., i.e.,*

$\left|u_{ci}\right|\leq u_{max}$

*.*

*The bound on the additive fault generally depends on the type of systems and conditions that they are working in. Therefore, it varies from one system to another, and it could be estimated based on our knowledge of the systems.*


**Assumption** **2.**
*Compound disturbances that are imposed to the system are bounded, i.e., always there is a constant parameter*

$d_{0}$

*where*

$\|d\|\leq d_{0}$

*.*


The dynamic of the leader is considered as: (8)$$D^{q}y=g\left(y\right),$$
where $y={\left[y_{1},\;y_{2},\ldots ,y_{n}\right]}^{T}.$

### 3.2. Control Design and Stability Analysis

We define the consensus protocol error as follows:(9)$$e_{i}=\sum_{j=1}^{N}a_{ij}\left(x_{i}-x_{j}\right)\;+\;b_{i}\left(x_{i}-y\right)\;=(\sum_{j=1}^{N}a_{ij}+b_{i})x_{i}-\sum_{j=1}^{N}a_{ij}x_{j}\;-\;b_{i}y,$$

By considering Equations (7) and (9), one can reach
(10)$${\dot{e}}_{i}=(\sum_{j=1}^{N}a_{ij}+b_{i}){\dot{x}}_{i}-\sum_{j=1}^{N}a_{ij}{\dot{x}}_{j}-b_{i}\dot{y}=(\sum_{j=1}^{N}a_{ij}+b_{i})\left(f\left(x\right)\;+\;u_{c}+N\right)-\;\sum_{j=1}^{N}a_{ij}{\dot{x}}_{j}-b_{i}\dot{y},$$
where
(11)$$N=B\left(t\right)\;\left(\left(E\;-\;I\right)u_{c}+\;\bar{u}\right)\;+\;d.$$

The following condition holds based on Assumptions 1 and 2:(12)$$N\left(d,u_{c}\right)\;\leq \Delta .$$

### 3.3. Super-Twisting SMC

Now, by applying a finite-time disturbance observer, we design the finite-time super-twisting TSMC for system (7). The sliding surface is defined as follows:(13)$$s_{i}\left(t\right)\;=\tau e_{i}\left(t\right),$$
in which τ is a positive user-defined constant. Finally, the control law of the proposed fault-tolerant and disturbance-observer-based finite-time super-twisting is given by:(14)$$\begin{array}{cc} u_{ci}= & \frac{1}{\left(\sum_{j=1}^{N}a_{ij}+b_{i}\right)}\left(\overset{N}{\underset{j=1}{\sum }}a_{ij}\left({\dot{x}}_{j}\right)+b_{i}\dot{y}\right)\\ & -\left(u_{si1}+\varsigma_{1}+f\left(x_{i}\right)\right)\\ u_{si1}= & -k_{1}{\left|s_{i}\right|}^{\frac{1}{2}}sign\left(s_{i}\right)+u_{si2}\\ {\dot{u}}_{si2}= & -k_{2}sign\left(s_{i}\right), \end{array}$$
where parameters $k_{1}$ and $k_{2}$ are positive user-defined parameters. Also, $\varsigma_{1}$ is the value of the estimated compound disturbance, and the following formulae are used to calculate it:(15)$$\begin{array}{cc} {\dot{\varsigma }}_{0}= & \xi_{0}+u_{c}+f_{i}\left(x\right)\;-\;g\left(y\right)\\ \xi_{0}= & -\alpha_{1}\;L^{\frac{1}{3}}\;{\left|\varsigma_{0}-e_{i}\right|}^{\frac{2}{3}}\mathrm{sign}\left(\varsigma_{0}-D^{q-1}e_{i}\right)\;+\;\varsigma_{1}\\ {\dot{\varsigma }}_{1}= & \xi_{1}\\ \xi_{1}= & -\alpha_{2}\;L^{\frac{1}{2}}\;{\left|\varsigma_{1}-\xi 0\right|}^{\frac{1}{2}}\mathrm{sign}\left(\varsigma_{1}-\xi_{0}\right)\;+\;\varsigma_{2}\\ {\dot{\varsigma }}_{2}= & -\alpha_{3}\;L\left|\varsigma_{2}-\xi 1\right|\;\mathrm{sign}\left(\varsigma_{2}\;-\;\xi_{1}\right), \end{array}$$
where, $\varsigma_{j}\;:={\left[\varsigma_{j1},\;\varsigma_{j2},\ldots ,\;\varsigma_{jn}\right]}^{T},\;\xi_{j}\;:={\left[\xi_{j1},\;\xi_{j2},\ldots ,\;\xi_{j3}\right]}^{T},\;j=0,\;1$ and, $\alpha_{1},\;\alpha_{2}\;\mathrm{and}\;\alpha_{3}>0,\;i=1,\;2,\;3$. Also, $L=\mathrm{diag}\left(L_{1},L_{2},\ldots ,\;L_{n}\right)>0$.

**Theorem** **1.**
*Under the control law (14) and finite time disturbance observer (15), the states of the follower systems converge to the desired value in finite time.*


**Proof.** At first, we prove that the estimator is able to precisely monitor compounded nonlinearity N. To this end, we define the auxiliary error variables as follows:


(16)
$$e_{\varsigma_{0}}=\varsigma_{0}-e_{i},\;e_{\varsigma_{1}}=\varsigma_{1}-N,\;e_{\varsigma_{2}}=\varsigma_{2}-\dot{N}.$$


Taking into account Equation (15), we have:(17)$${\dot{e}}_{\varsigma_{0}}=-\alpha_{1}\;L^{\frac{1}{3}}\;{\left|\varsigma_{0}-e\right|}^{\frac{2}{3}}sign\left(\varsigma_{0}-e\right)+\varsigma_{1}-N\;=-\alpha_{1}\;L^{\frac{1}{3}}\;{\left|\varsigma_{0}-e\right|}^{\frac{2}{3}}sign\left(e_{\varsigma_{0}}\right)+e_{\varsigma_{1}}\;{\dot{e}}_{\varsigma_{1}}=-\alpha_{2}\;L^{\frac{1}{2}}\;{\left|\varsigma_{1}-\xi 0\right|}^{\frac{1}{2}}sign\left(\varsigma_{1}-\xi_{0}\;\right)+\varsigma_{2}-N\;=-\alpha_{2}\;L^{\frac{1}{3}}\;{\left|e_{\varsigma_{1}}-{\dot{e}}_{\varsigma_{0}}\right|}^{\frac{1}{2}}sign\left(e_{\varsigma_{1}}-{\dot{e}}_{\varsigma_{0}}\right)+e_{\varsigma_{2}}\;{\dot{e}}_{\varsigma_{2}}=-\alpha_{3}\;L\left|\;\varsigma_{2}-\xi 1\;\right|\;sign\left(\varsigma_{2}-\xi 1\right)-\ddot{N}\;=-\alpha_{3}\;L\left|e_{\varsigma_{2}}-{\dot{e}}_{\varsigma_{1}}\right|\;sign\left(e_{\varsigma_{2}}-{\dot{e}}_{\varsigma_{1}}\right)-\ddot{N}$$
that is,
(18)$$\begin{array}{cc} {\dot{e}}_{\varsigma_{0}i}= & -\alpha_{1}\;l_{i}^{\frac{1}{3}}\;{\left|e_{\varsigma_{0}i}\;\right|}^{\frac{2}{3}}sign\left(e_{\varsigma_{0}i}\right)+e_{\varsigma_{1}i}\\ {\dot{e}}_{\varsigma_{1}i}= & -\alpha_{2}\;l_{i}^{\frac{1}{2}}\;{\left|e_{\varsigma_{1}i}-e_{\varsigma_{0}i}\;\right|}^{\frac{1}{2}}sign\left(e_{\varsigma_{1}i}-{\dot{e}}_{\varsigma_{0}i}\right)+e_{\varsigma_{2}i}\\ {\dot{e}}_{\varsigma_{2}i}\;\in & -\alpha_{3}\;l_{i}\left|\;e_{\varsigma_{2}i}-e_{\varsigma_{1}i}\right|\;sign\left(e_{\varsigma_{2}i}-{\dot{e}}_{\varsigma_{1}i}\right)+\left[-L_{N},\;L_{N}\right]. \end{array}$$

According to Lemma 2 in reference [58], it can be established that in a finite time, the approximation errors $e_{\varsigma_{0}i},\;e_{\varsigma_{1}i}$ and $e_{\varsigma_{2}i}$ converge to zero. Hence, after $0<T_{o}<\infty$ the following equations hold
(19)$$\varsigma_{0}\;\left(t\right)=e_{i},\;\varsigma_{1}\;\left(t\right)=N\left(t\right),\;\varsigma_{2}\;\left(t\right)=\dot{N}.$$

Now, we prove the finite-time stability of the closed-loop system. Substituting the proposed control law (14) in the sliding surface’s time-derivative results in:
(20)$${\dot{s}}_{i}=\tau {\dot{e}}_{i}=\tau \left(\overset{N}{\underset{j=1}{\sum }}a_{ij}+b_{i}\right)\left(f\left(x\right)+u_{c}+N\right)-\tau \overset{N}{\underset{j=1}{\sum }}a_{ij}{\dot{x}}_{j}\;-b_{i}\dot{y}=\tau \left(\overset{N}{\underset{j=1}{\sum }}a_{ij}+b_{i}\right)(f\left(x\right)\;+\frac{1}{\left(\sum_{j=1}^{N}a_{ij}+b_{i}\right)}\left(\overset{N}{\underset{j=1}{\sum }}a_{ij}\left({\dot{x}}_{j}\right)+b_{i}\dot{y}\right)-\left(u_{si1}+\varsigma_{1}+f\left(x_{i}\right)\right)+N)\;-\tau \overset{N}{\underset{j=1}{\sum }}a_{ij}{\dot{x}}_{j}-b_{i}\dot{y}=\tau \left(\overset{N}{\underset{j=1}{\sum }}a_{ij}+b_{i}\right)\left(-u_{si1}+N-\varsigma_{1}\right)$$

According to Equation (12), $N=\varsigma_{1}$, consequently, we have: (21)$${\dot{s}}_{i}=-\tau (\sum_{j=1}^{N}a_{ij}+b_{i})\;u_{si1}.$$

As a result, we have: (22)$$\begin{array}{cc} {\dot{s}}_{i}= & -\tau \left(\overset{N}{\underset{j=1}{\sum }}a_{ij}+b_{i}\right)\;k_{1}{\left|s_{i}\right|}^{\frac{1}{2}}sign\left(s_{i}\right)+u_{si2}\\ & {\dot{u}}_{si2}=-k_{2}sign\left(s_{i}\right). \end{array}$$

By defining new variables $w_{1}=s_{i}$ and $w_{2}=u_{si2}$ and rearranging the equation above, we get:(23)$$\begin{array}{cc} {\dot{w}}_{1}= & -\tau \left(\overset{N}{\underset{j=1}{\sum }}a_{ij}+b_{i}\right)\;k_{1}{\left|w_{1}\right|}^{\frac{1}{2}}sign\left(w_{1}\right)+w_{2}\\ {\dot{w}}_{2}= & -k_{2}sign\left(w_{1}\right), \end{array}$$
in which Equation (23) represents a second-order super-twisting algorithm. On the basis of Theorem 1 [59] and its proof, the following Lyapunov function is considered:(24)$$V_{0}=\varsigma^{T}P\varsigma ,$$
where $\varsigma ={\left[\varsigma_{1},\varsigma_{2}\right]}^{T}={\left[{\left|w_{1}\right|}^{\frac{1}{2}}sign\left(w_{1}\right),w_{2}\right]}^{T}$. $V_{0}$ is quadratic, robust and strict with symmetric and positive definite matrix P will fulfil: (25)$${\dot{V}}_{0}=-{\left|w_{1}\right|}^{\frac{1}{2}}\varsigma^{T}Q\varsigma$$
for symmetric and positive definite matrix *Q*. Furthermore, the trajectory starting at $w\left(0\right)$ will arrive at the origin at $t_{f}$, which is given by:(26)$$t_{f}=t_{s\Delta }+\frac{2\lambda_{max}\left\{P\right\}}{\lambda_{min}^{\frac{1}{2}}\left\{P\right\}\lambda_{min}\left\{Q\right\}\;}V_{0}^{\frac{1}{2}}\left(t_{0}\right).$$

The Lyapunov function’s matrices *P* and Q can be chosen using the technique outlined in [59], which ensures that the sliding variables $w_{1}$ and $w_{2}$ reach zero in a specified amount of time. □

## 4. Results

For the simulations, the parameters of leader and all agents are the same and are equal to what was mentioned in Section 2, while different initial conditions are considered for each follower agent. Both bias faults and partial loss of effectiveness are taken into account when evaluating the performance of the suggested control techniques. The structure of these faults is considered based on Equation (4) and Table 1.

The external disturbances are also considered as follows:(27)$$d_{i}=4\sin\left(\frac{j\pi }{2}t\right)-2\cos\left(j\pi t\right).$$

### 4.1. Simulation without Active Control Constraint

The control scheme is turned on at t=6. The time history of distributed consensus tracking is illustrated in Figure 4. Also, in Figure 5, the errors of distributed consensus tracking are shown. As demonstrated in these figures, all agents follow the leader and distributed consensus tracking is achieved after a finite time. The values of control commands are displayed in Figure 6. As it is shown, thanks to the proposed super-twisting SMC, there is no chattering in the numerical results. Also, Figure 7 shows the perfect agreement between the time history of estimated and actual values of the actuator faults and external disturbances, demonstrating the excellent performance of the proposed observer.

### 4.2. Simulation with Active Control Constraint

Now, we repeat the simulations with active constraints on the control inputs. In real-world life systems, to avoid large control inputs, we must select an appropriate limitation on control inputs. Nonetheless, in most studies in this field, this matter is completely ignored. Herein, we consider the following bounds for control input of all followers:(28)$$\max\left(\left|u_{1}\right|\right)\;\leq 150,\;\max\left(\left|u_{2}\right|\right)\;\leq 150\max\left(\left|u_{3}\right|\right)\;\leq 150.$$

Figure 8, Figure 9, Figure 10 and Figure 11 show the numerical results under the proposed technique when the followers are in the presence of control input constraints. As is shown in these figures, the control acts well even when we impose these constraints on it.

## 5. Discussions

Though in the literature there are several promising methods for consensus tracking control of chaotic multi-agent systems, there is still room for improvement of existing methods. The aforementioned concerns have prompted the current study. Furthermore, the majority of consensus tracking control methods have significant flaws that make their real-world implementation difficult. Hence, this study proposes a finite-time and chatter-free approach for distributed consensus tracking of chaotic multi-agent supply chain networks. The super-twisting method is used to develop an effective controller without chattering. To reduce the negative effects of uncertainty and interruptions, the proposed method includes a finite-time disturbance observer. In the presence of disturbances and control input faults, the distributed consensus tracking of chaotic multi-agent supply chain networks is studied. Finally, the effectiveness of the suggested control strategy was evaluated using numerical simulations.

Because of their advantages, such as guaranteed stability, robustness against parameter changes, and ease of implementation, SMC and adaptive control approaches have received a lot of attention among researchers in the control field [60,61,62,63,64]. Nonetheless, the main issue with the SMC is the chattering phenomenon caused by the discontinuous functions [65]. The current investigation was prompted by this concern. As it was shown through numerical results. The proposed control technique is suitable for distributed consensus tracking of nonlinear uncertain systems. In the proposed scheme, the effects of faults in control actuators and robustness against unknown time-varying disturbances are taken into account. The proposed technique also uses a finite-time super-twisting algorithm that avoids chattering in the system’s response and control input.

As shown in Figure 4, the leader system is perfectly tracked by all agents within a short amount of time, and the distributed consensus tracking goal is completely achieved, demonstrating the proper performance of the suggested control approach. The proposed method’s superior performance is due to the outstanding performance of the proposed adaptive mechanism, which precisely estimates the slave system’s uncertain parameters (see Figure 7). Based on our numerical analysis, after t=8 the error of estimation and distributed consensus tracking control are less than 2% and 1%, respectively. Hence, in practical applications for the control and synchronization of real-world supply chains, the proposed methodology is able to meet the expected performance even when there are various kinds of actuator faults and disturbances.

Moreover, by comparing the results of the systems with and without control constraint, as expected, the system is faster without control constraint. However, in practical systems, control constraints should be considered according to the physical and instrumental limitations of actuators. One of the main advantages of the offered distributed consensus control technique in this study is tracking control even in the presence of control input constraints.

## 6. Conclusions

In this study, the distributed consensus tracking of chaotic multi-agent supply chain networks through the new fault-tolerant, finite-time, and chatter-free approach was studied. At first, the model of a chaotic supply chain network was presented. The supply chain network’s chaotic response was exhibited. Then, a multi-agent system based on the supply chain network was offered. After that, a new control scheme for distributed consensus tracking of the system was proposed, and its design procedure was delineated. In comparison with its state-of-the-art counterparts, the proposed methodology makes the multi-agent systems robust against faults and failures in control input, as well as uncertainties and external. Furthermore, the suggested control ensures finite-time performance, and by means of the super-twisting algorithm, it provides smooth responses. The Lyapunov stability theorem was used to show the system’s finite-time convergence and stability. Finally, the suggested scheme’s luminous performance was demonstrated through numerical simulations. For instance, for the system investigated in this research, it is shown that after less than two units of time, all followers mimic the behavior of the leader, and the control purpose is fully achieved. At t=6, the controller and observer are turned on. After two units of time, the error of estimation is less than 2%. It remains in this bound (2 percent) forever. As a result, the controller receives accurate information and produces excellent results, that is, when the system is regulated for all followers, the error of distributed consensus tracking control is less than 1%. As a feature suggestion, since control input saturations have destructive effects on the performance of multi-agent systems, the proposed scheme can be promoted by taking advantage of a robust approach against control input saturation.

## Figures and Tables

**Figure 1 entropy-24-00033-f001:**
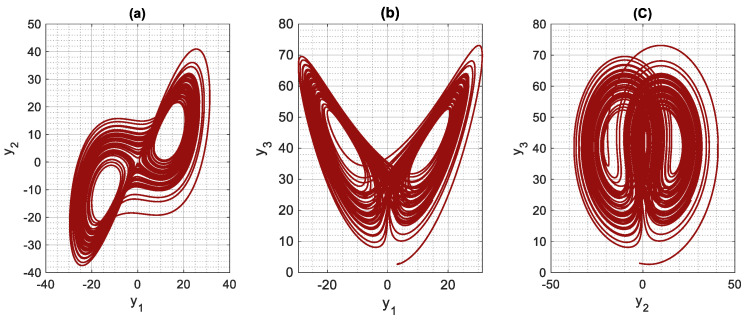
Two-dimensional (2D) phase plot of the chaotic supply chain network in the (**a**) $y_{1}-y_{2}$ plane, (**b**) $y_{1}-y_{3}$ plane, (**c**) $y_{3}-y_{3}$ plane.

**Figure 2 entropy-24-00033-f002:**
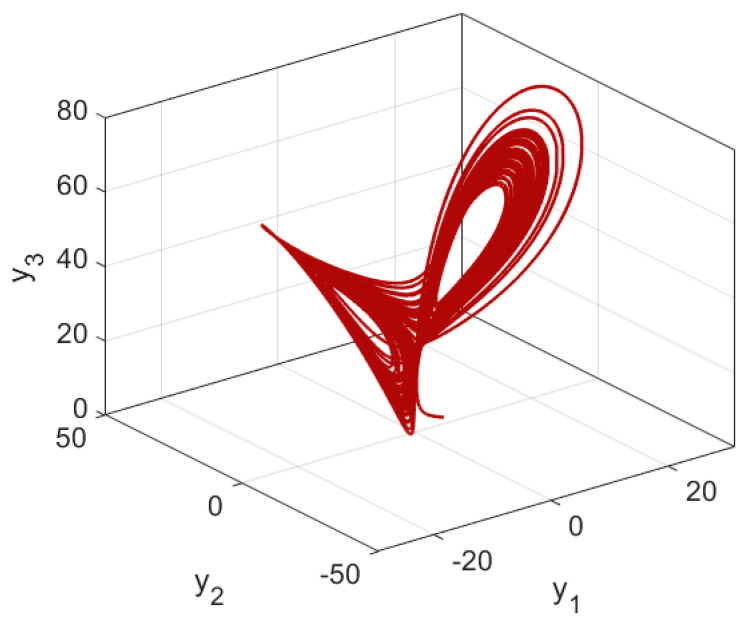
Three-dimensional (3D) phase plots of the chaotic supply chain network.

**Figure 3 entropy-24-00033-f003:**
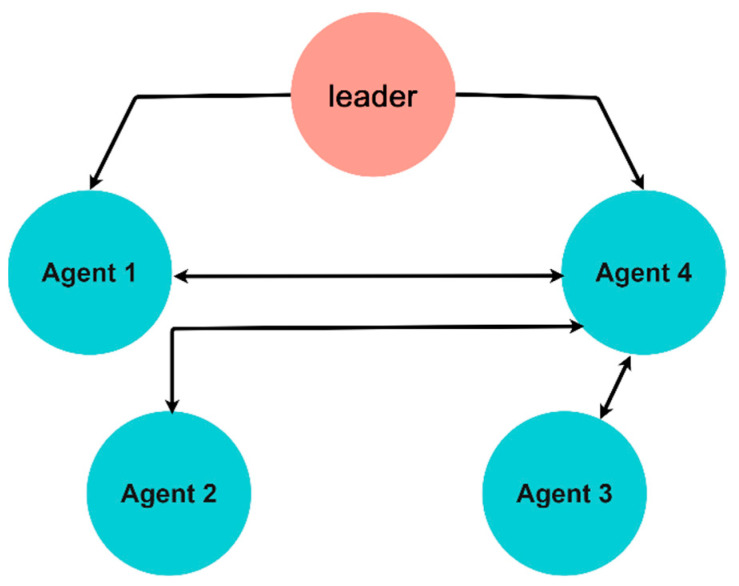
Communication topology of the multi-agent platform.

**Figure 4 entropy-24-00033-f004:**
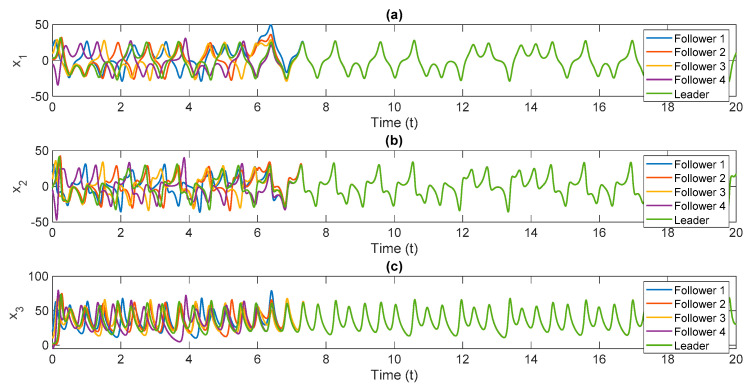
Simulation results of distributed consensus tracking of chaotic multi-agent supply chain networks in the presence of external disturbances, partial loss of effectiveness of actuator, and bias faults, (**a**) $x_{1}$ versus normalized time, (**b**) $x_{2}$ versus normalized time, (**c**) $x_{3}$ versus normalized time.

**Figure 5 entropy-24-00033-f005:**
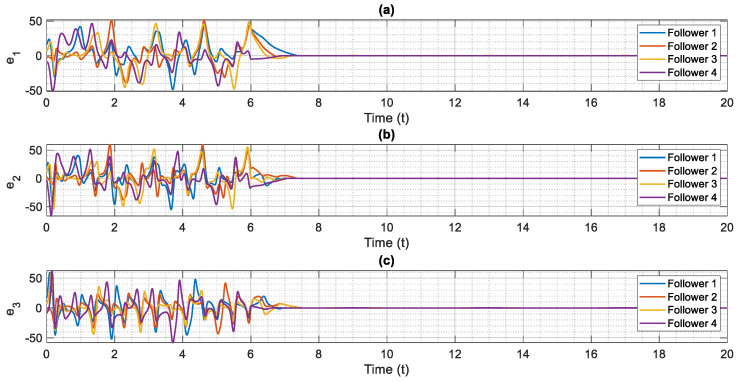
Error of distributed consensus tracking of chaotic multi-agent supply chain networks in the presence of external disturbances, partial loss of effectiveness of actuator and bias faults, (**a**) $e_{1}$ versus normalized time, (**b**) $e_{2}$ versus normalized time, (**c**) $e_{3}$ versus normalized time.

**Figure 6 entropy-24-00033-f006:**
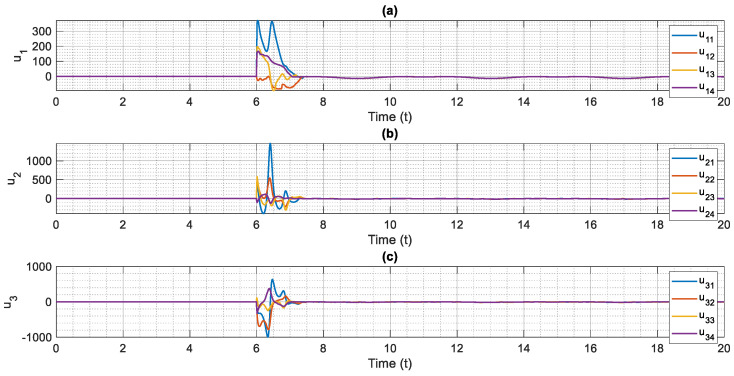
Control inputs based on the proposed fault-tolerant, finite-time and chatter-free super-twisting control technique, (**a**) $u_{1}$ versus normalized time, (**b**) $u_{2}$ versus normalized time, (**c**) $u_{3}$ versus normalized time.

**Figure 7 entropy-24-00033-f007:**
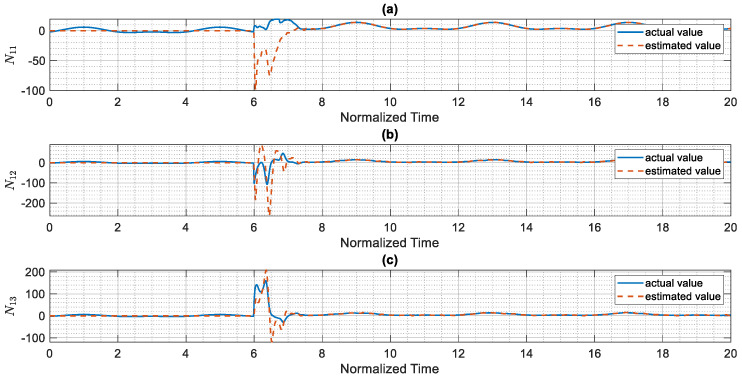
Results of the finite-time estimator control technique, (**a**) $N_{11}$ versus normalized time, (**b**) $N_{12}$ versus normalized time, (**c**)$N_{13}$ versus normalized time.

**Figure 8 entropy-24-00033-f008:**
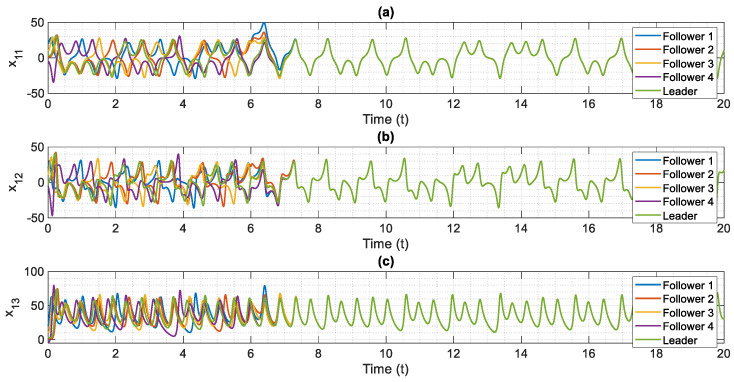
Simulation results of distributed consensus tracking of chaotic multi-agent supply chain networks in the presence of external disturbances, partial loss of effectiveness of actuator, bias faults, and active control constraint, (**a**) $x_{1}$ versus normalized time, (**b**) $x_{2}$ versus normalized time, (**c**) $x_{3}$ versus normalized time.

**Figure 9 entropy-24-00033-f009:**
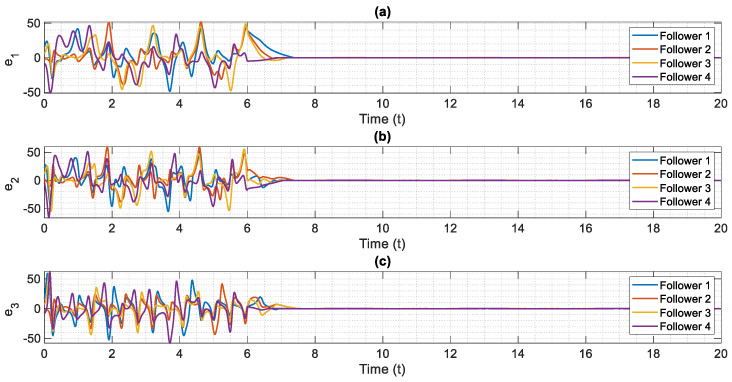
Error of distributed consensus tracking of chaotic multi-agent supply chain networks in the presence of external disturbances, partial loss of effectiveness of actuator, bias faults, and active control constraint, (**a**) $e_{1}$ versus normalized time, (**b**) $e_{2}$ versus normalized time, (**c**) $e_{3}$ versus normalized time.

**Figure 10 entropy-24-00033-f010:**
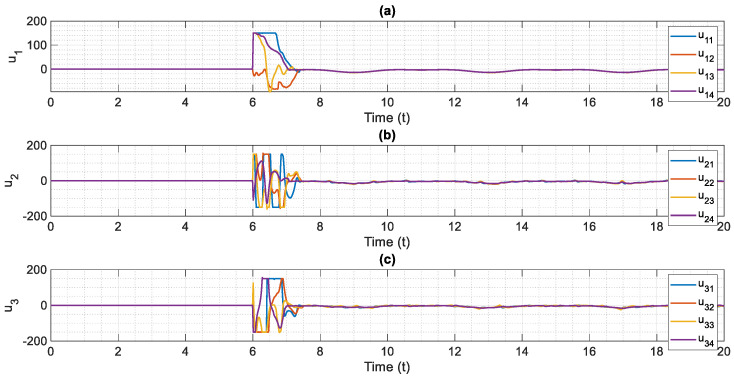
Control inputs based on the proposed fault-tolerant, finite-time and chatter-free super-twisting control technique in the presence of external disturbances, partial loss of effectiveness of actuator, bias faults, and active control constraint, (**a**) $u_{1}$ versus normalized time, (**b**) $u_{2}$ versus normalized time, (**c**) $u_{3}$ versus normalized time.

**Figure 11 entropy-24-00033-f011:**
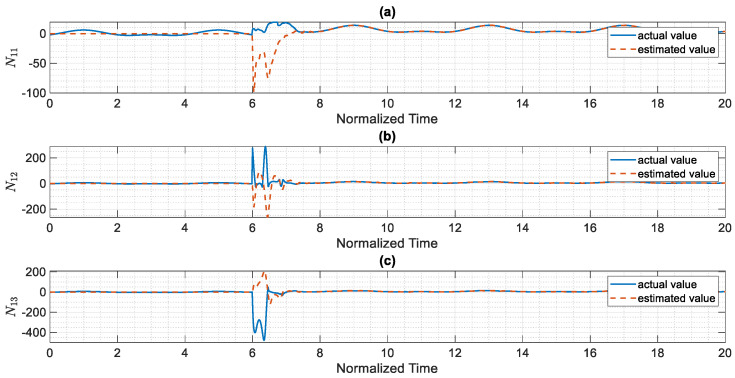
Results of the finite-time estimator control technique in the presence of external disturbances, partial loss of effectiveness of actuator, bias faults, and active control constraint, (**a**) $N_{11}$ versus normalized time, (**b**) $N_{12}$ versus normalized time, (**c**)$N_{13}$ versus normalized time.

**Table 1 entropy-24-00033-t001:** Parameters of actuator faults.

Parameters of the Faults	Value
Uncertain fault input $\left(\bar{u}\right)$	(5,5,5)
Fault evolution rate $\left(a_{i}\right)$	(12,12,12)
Actuator control effectiveness $\left(e_{ii}\right)$	(0.7,0.7,0.7)
Time of occurrence of the fault $\left(t_{0i}\right)$	(5,5,5)
Control constraint $\left(u_{max}\right)$	(150,150,150)

## Data Availability

Not Applicable.

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
