# Peer review of "Distributed Consensus Tracking Control of Chaotic Multi-Agent Supply Chain Network: A New Fault-Tolerant, Finite-Time, and Chatter-Free Approach"

_entropy, 2021, doi:10.3390/e24010033_

Round 1

Reviewer 1 Report

The authors propose a fault-tollerant approach to consensus tracking control, taking actuator or sensor faults into account, what suggest nice implementation properties of the proposed approach. 

The supply chain vunerability and its robust properties are of prime importance, as far as recent events are taken into consideration, starting from a shortage of semiconductors, to oversized consumption expectation towards certain goods, which are unable to be satisfied due to a corona-related crisis.

In the introductory part of the paper, the authors identify various control scenarios which have inherent robustifying features, such as the sliding-mode control. I would suggest the authors to rephrase the ending paragraphs of their introduction to give clear statements, even using bullet lists, comprising the novelty and contribution separately. 

Instead of Section 2.2 I would rather see some formal definitions, as Definition 2.1 (weighed graph), Definition 2.2 (directed graph), etc

Please remove typo from Fig. 3. 

No time indices are given in (1) or (2), nor is there anything mentioned about omitting them, thus the notation dot{x} seems strange and unsupported. 

Are there any assumptions concerning the uncertainties in (3)? Bounds? Satisfaction of a small-gain theorem? 

How can one identify/define the bound on the additive fault in Assumption 1 a priori? 

In order to understand the proof of Theorem 1, the authors should, at first, undergo a typesetting task, to put all the expressions in a readable format, without ejecting lines. 

Please mention the bounds used in the algoritm, in Table 1. 

Figure 6 - any control limits taken into account? Are the amplitudes large or small here? What if a control signal got saturated? Can you present a transient behaviour of your system subject to the same configuration and initial conditions while one of the controls is saturated? 

This is at the same time the major drawback of the paper, as initially the authors have claimed the sytstem has robust properties, especially against actuator fault. 

This needs a better support. 

Author Response

Reviewer 1

The authors propose a fault-tollerant approach to consensus tracking control, taking actuator or sensor faults into account, what suggest nice implementation properties of the proposed approach. 

The supply chain vunerability and its robust properties are of prime importance, as far as recent events are taken into consideration, starting from a shortage of semiconductors, to oversized consumption expectation towards certain goods, which are unable to be satisfied due to a corona-related crisis.

  1. In the introductory part of the paper, the authors identify various control scenarios which have inherent robustifying features, such as the sliding-mode control. I would suggest the authors to rephrase the ending paragraphs of their introduction to give clear statements, even using bullet lists, comprising the novelty and contribution separately. 

The authors thank the reviewer for this appropriate comment. Based on this comment, the introduction section is modified, and bullet lists have been added.

  1. Instead of Section 2.2 I would rather see some formal definitions, as Definition 2.1 (weighed graph), Definition 2.2 (directed graph), etc

According to this valuable comment, this subsection has been replaced with two definitions.

  1. Please remove typo from Fig. 3. 

Thank you very much for the great point you mentioned. Figure 3 has been modified, and the typo has been corrected.

  1. No time indices are given in (1) or (2), nor is there anything mentioned about omitting them, thus the notation dot{x} seems strange and unsupported. 

Many thanks for your consideration. The states of the system are time-dependent. Based on this comment, we have corrected equations 1 and 2.

  1. Are there any assumptions concerning the uncertainties in (3)? Bounds? Satisfaction of a small-gain theorem? 

Thank you for your comment. In Eq. (3), we only define a generalized form of the system, and we do not mention the bounds and Satisfaction of a small-gain theorem. But for the design of the controller, these criteria are defined and mentioned.

  1. How can one identify/define the bound on the additive fault in Assumption 1 a priori? 

We really appreciate your consideration. The bound on the additive fault generally depends on the type of systems and conditions that they are working in. So basically, it varies from one system to another, and it could be estimated based on our knowledge of the systems.

Based on this comment, some descriptions have been added to the paper after assumption 1. 

  1. In order to understand the proof of Theorem 1, the authors should, at first, undergo a typesetting task, to put all the expressions in a readable format, without ejecting lines. 

Thank you for the suggestion; we modified the proof of Theorem 1.

  1. Please mention the bounds used in the algorithm, in Table 1. 

Based on this comment, the used bounds have been added to Table 1.

  1. Figure 6 - any control limits taken into account? Are the amplitudes large or small here? What if a control signal got saturated? Can you present a transient behaviour of your system subject to the same configuration and initial conditions while one of the controls is saturated? 

Thank you for this suggestion. In this paper, we did not consider the effects of saturation on the system. In a future study, we aim to, design a robust controller again control input saturation and investigate all effects of saturation, and try to eliminate the destructive effects on the system.

This has been added to the paper as a future suggestion.

  1. This is at the same time the major drawback of the paper, as initially the authors have claimed the system has robust properties, especially against actuator fault. 

This needs a better support.

Thank you very much for your comment. To address this concern, we have added a discussion section and investigated the features of the proposed method.

Reviewer 2 Report

Review of the article „Distributed consensus tracking control of chaotic multi‐agent supply chain network: A new fault‐tolerant, finite‐time, and chatter‐free approach“, authors: Ziyi Liu, Hadi Jahanshahi, Christos Volos, Stelios Bekiros, Shaobo He, Madini O. Alassafi, Adil M. Ah‐mad

Shortcomings of the article:

The aim of the research needs to be clearly stated and presented at the end of the introduction.

It is not clear how the primary information was obtained, how it was selected and what research methodology was used.

In figures 1 and from 4 to 7 need marked graphs a, b, c and them explanations. The everything needs to be increased. The units of axes Y1, Y2 and Y3 must be specified.

In figure 2 the units of axes Y1, Y2 and Y3 must be indicated.

Needs add research results section.

Needs add discussions section. In this section need to discuss about research results.

The practical benefits of the research must be clearly demonstrated also needs add information about the research application possibilities in practice.

In the conclusions must clearly show what problems the researchers have solved and how much to get results are better than the results of other researches. The conclusions should be clear and concise with the numerical values provided to support and justify the results obtained. The presented conclusions are not informative. Conclusions need to be rewritten.

The relevance of the article is obvious, but this article is more like a report on the work done, but not a scientific article. The article needs to be redesigned to make clear the results and their practical benefits.

Author Response

Reviewer 2

Shortcomings of the article:

  1. The aim of the research needs to be clearly stated and presented at the end of the introduction.

We appreciate the reviewer’s concern. Based on this comment, the introduction section is modified, and the motivation of the system has been added much more clearly.

  1. It is not clear how the primary information was obtained, how it was selected and what research methodology was used.

The authors thank the reviewer for this valuable consideration. Accordingly, the description of primary information and research methodology has been added to the paper in section 2.2.

  1. In figures 1 and from 4 to 7 need marked graphs a, b, c and them explanations. The everything needs to be increased. The units of axes Y1, Y2 and Y3 must be specified.

Thank you. Based on this comment, we modified these figures.

  1. In figure 2 the units of axes Y1, Y2, and Y3 must be indicated.

Thank you for your comment. Since we have used the non-dimensional model in this study, Y1, Y2 and Y3 are dimensionless. This has been mentioned in the revised paper.

  1. Needs add research results section.

Thank you for this valuable comment. Based on this comment, the structure of the paper has been changed

  1. Needs add discussions section. In this section need to discuss about research results.

Thank you for this valuable comment. Accordingly, the discussion section has been added.

  1. The practical benefits of the research must be clearly demonstrated also needs add information about the research application possibilities in practice.

Thank you for your great comment. In the discussion section of the revised paper, we have elaborated on the practical benefits of the current study.

  1. In the conclusions must clearly show what problems the researchers have solved and how much to get results are better than the results of other researches. The conclusions should be clear and concise with the numerical values provided to support and justify the results obtained. The presented conclusions are not informative. Conclusions need to be rewritten.

According to this comment, the conclusion has been modified.

  1. The relevance of the article is obvious, but this article is more like a report on the work done, but not a scientific article. The article needs to be redesigned to make clear the results and their practical benefits.

We appreciate your concern; we revised most parts of the manuscript to meet the expected criteria.

Round 2

Reviewer 1 Report

Thank you for taking my comments into account, however I still believe that the simulation with constraints, and possible comment should be included in the paper. Also please consider removing <CR>s from the proof of Theorem 3.2. 

Author Response

Reviewer#1-Round#2

Thank you for taking my comments into account, however I still believe that the simulation with constraints, and possible comment should be included in the paper. Also please consider removing <CR>s from the proof of Theorem 3.2.

The authors thank the reviewer for the useful comments. Based on this comment, again, we tried to modify the paper. We have arranged the formulas in proof theorem 3.2. for the second revise, and we hope we have followed your suggestion.  Also, many thanks for your help and suggestion; we are working on a controller that is equipped with a finite-time sliding mode mechanism, and the effects of constraints are considered in our next study.

Reviewer 2 Report

In figure 1 need marked graphs a, b, c and them explanations.

In figures 1, 2, 4, 5, 6 and 7 the measurements values of coordinates axes must be indicated.

The conclusions should be clear and concise with the numerical values provided to support and justify the results obtained.

Author Response

Reviewer#2-Round#2

In figure 1 need marked graphs a, b, c and them explanations. In figures 1, 2, 4, 5, 6 and 7 the measurements values of coordinates axes must be indicated. The conclusions should be clear and concise with the numerical values provided to support and justify the results obtained.

Thank you for your consideration. Figure 1 has been modified, and explanations of a, b, c have been added. Also, in 1, 2, 4, 5, 6, and 7, the measurements values of coordinates axes have been added (we have added dash lines that show the measurement for the axes, and we hope we have followed your suggestion). In addition, we added some points about numerical values (the percent of estimation and control error) in conclusion.

Round 3

Reviewer 1 Report

Thank you for incorporating the requested changes into the introductory part of the paper, however, as far as limits are concerned - I do not ask you to include a new research, I need a result to be presented where ACTIVE CONSTRAINTS are presented in the simulation case, not the constraints at the level of 2000 which remain inactive. A short discussion following the presented results should improve the paper. 

Author Response

We really appreciate your concern, accordingly in the revised paper, we have added new results (highlighted in Turquoise) considering the effects of active control input constraints.

Round 4

Reviewer 1 Report

Thank you for introducing Section 4.2. The results presented are interesint, but just please re-write the sentence considering SELECTION of appropriate limitations - these result from constraints present in the process, not from our own actions to clip control signals!